



# The physics behind Van der Burgh's empirical equation, providing a new predictive equation for salinity intrusion in estuaries

Zhilin Zhang[1] and Hubert. H. G. Savenije[1]

[1]Department of Water Management, Faculty of Civil Engineering and Geosciences, Delft University of Technology, Delft, the Netherlands

*Correspondence to:* Zhilin Zhang (z.zhang-5@tudelft.nl)

**Abstract.**

The practical value of the surprisingly simple Van der Burgh equation to predict saline water intrusion in alluvial estuaries is well documented, but the physical foundation of the equation is still weak. In this paper we provide a connection between the empirical equation and the theoretical literature, leading to a theoretical range for Van der Burgh's coefficient of $1/2 < K < 2/3$ which falls within the feasible range of $0 < K < 1$. In addition, we developed a one-dimensional predictive equation for the dispersion of salinity as a function of local hydraulic parameters that can vary along the estuary axis, including mixing due to residual circulation. This type of mixing is relevant in the wider part of alluvial estuaries where preferential ebb and flood channels appear. Subsequently, this dispersion equation is combined with the salt balance equation to obtain a new predictive analytical equation for the longitudinal salinity distribution. Finally, the new equation was tested and applied to a large database of observations in alluvial estuaries, whereby the calibrated $K$ values appeared to correspond well with the theoretical range.

## 1 Introduction

Estuaries play an essential role in the human-earth system, affecting fresh water resources, the mixing between ocean and river water, and the health of aquatic ecosystems. This makes the functioning of estuarine systems an important field of research. A crucial element of estuarine dynamics is the interaction between saline and fresh water. The river discharges fresh water into estuaries, flushing out the salt, while saline water penetrates landward as a result of density gradients. The temporal and spatial distribution of salinity in an estuary is determined by the competition between fresh water flushing and penetration of saline water by gravity.

Dispersion is the mathematical reflection of the spreading of a substance (e.g., salinity $s$) through a fluid as a function of a gradient in the concentration of the substance (e.g., the salinity gradient $\mathrm{d}s/\mathrm{d}x$). Hence, dispersion is the mathematical description of mixing. The physical process driving dispersion differs at different scales, depending on the dominant mechanism. For instance, at the molecular scale, the dominant mechanism is the Brownian movement of water molecules. At the scale of river flow, the process is driven by the transfer of friction from the riverbed into the cross section through turbulence. At this scale, the dispersion coefficient is called hydraulic eddy viscosity ($K_E$) (Dyer, 1997). The most important type of mixing in estuaries is the result of salinity gradients and the non-concurrence of the velocity and salinity field ($\overline{u's'}$) (MacCready, 2004), which is



the result of gravitational and tidal mixing processes. Finally, there is mixing by residual circulation, driven by the tide, where ebb and flood flows of different densities mix (e.g., Nguyen et al., 2008).

The dispersion resulting from density gradients is closely connected to the stratification number $N_R$, which is the balance between the potential energy resulting from the buoyancy of fresh water flowing into the estuary and the kinetic energy of

the tide that provides the energy of mixing. This stratification number, also known as the Estuarine Richardson Number, is widely used in theoretical and practical studies (e.g., Fischer, 1972; Savenije, 1986; Kuijper and Van Rijn, 2011). If $N_R$ is large, potential energy of river discharge dominates and stratification occurs; if $N_R$ is small, the estuary is well mixed due to sufficient kinetic energy to reduce the density gradient.

Van der Burgh (1972) developed a purely empirical method with excellent practical performance (e.g., Savenije, 2005), com-

bining into one equation the effects of all mixing mechanisms. However, the physical meaning of Van der Burgh's coefficient $K$ is still unknown. Starting from this equation, the dispersion coefficient $D$ can be shown to be proportional to the salinity gradient to the power of $K/(1-K)$ (Savenije, 2015). The literature presents different values for this power. Transferring these back with this relationship to Van der Burgh's coefficient, we found a theoretical value of 1/2 (Fischer, 1976; Thatcher and Najarian, 1983; Kuijper and Van Rijn, 2011), of 1 (Hansen and Rattray, 1965), a series of 0, 1/2 and 2/3 (Prandle, 1981; Mac-

Cready, 2004) or an empirical range of 0.20-0.75 (Gisen, 2015a). This article aims to provide a theoretical background for this coefficient.

Traditionally, researchers focused on vertical/longitudinal dispersion in prismatic estuaries (Hansen and Rattray, 1965) or cross-sectional varying estuaries (Prandle, 1981; MacCready, 2004). Fischer (1972) concluded that the lateral gravitational circulation is dominant over the sum of vertical oscillatory shear, net vertical circulation and lateral oscillatory shear. Lerczak

and Geyer (2004) also stated the importance of lateral circulation to the momentum budget in estuaries but they used straight and prismatic channels, whereas the fact that the cross sections of natural alluvial estuaries obey an exponential function is relevant. In addition, almost all researchers split-up dispersion into its components by decomposed salinity and velocity (e.g., Hansen and Rattray, 1965; Fischer, 1972; Prandle, 1981; Thatcher and Najarian, 1983; MacCready, 2004, 2007, 2011; Lerczak and Geyer, 2004; Ralston and Stacey, 2005). Moreover, several researchers determined the dispersion based on a downstream

boundary (Hansen and Rattray, 1965; Kuijper and Van Rijn, 2011; Gisen et al., 2015b), instead of calculating local dispersion on the basis of local hydraulic variables, as done in this research.

Although the processes of mixing and saline water intrusion are clearly complex and three-dimensional, it is remarkable that a very simple, empirical and one-dimensional approach, such as Van der Burgh's relationship, has yielded such surprisingly good results. This paper tries to bridge the gap between the theoretical approaches developed in the literature and the empirical

results obtained with Van der Burgh's relationship, considering the complex interaction between tide, geometry, salinity and fresh water that govern dispersion in alluvial estuaries. In addition, we present a one-dimensional general dispersion equation for convergent estuaries that includes lateral exchange through preferential ebb and flood channels, using local tidal and geometrical parameters. This equation was validated on a broad database of salinity distributions in alluvial estuaries.



## 2  Linking Van der Burgh to the Traditional Literature

The one-dimensional mass-conservation equation averaged over the cross section over a tidal cycle can be written as (e.g., Savenije, 2005):

$$A\frac{\partial s}{\partial t} - |Q_f|\frac{\partial s}{\partial x} - \frac{\partial}{\partial x}\left(DA\frac{\partial s}{\partial x}\right) = 0 \tag{1}$$

where $A = Bh$ is the cross-sectional area, $B$ is the width, $h$ is the depth, $s$ is the cross-sectional average salinity, $t$ is time, $Q_f$

is the fresh water discharge, $x$ is the distance from estuary mouth and $D$ is the effective longitudinal dispersion coefficient. The positive direction of flow is in the upstream direction.

At steady state, where $\partial s/\partial t = 0$, using the boundary condition: at $x \to \infty$, $s = s_f$ and $\partial s/\partial x = 0$, integration yields:

$$-\frac{|Q_f|}{A}(s - s_f) = D\frac{\mathrm{d}s}{\mathrm{d}x} \tag{2}$$

where $s_f$ is the fresh water salinity, usually close to zero.

Van der Burgh (1972) found an empirical equation describing the tidal average longitudinal variation of the effective dispersion:

$$\frac{\mathrm{d}D}{\mathrm{d}x} = -K\frac{|Q_f|}{A} \tag{3}$$

where the dimensionless coefficient $K \in (0, 1)$ according to Savenije (2005).

Combining equations (2) and (3) yields (Savenije, 1986, 1989, 1993a, b):

$$\frac{D}{D_1} = \left(\frac{s}{s_1}\right)^K \tag{4}$$

where $D_1$ and $s_1$ are the dispersion coefficient and salinity at a given point $x_1$, generally taken at the inflection point in the exponential estuary geometry. This equation is special in that it links the dispersion to the salinity, instead of the salinity gradient as most other researchers do (e.g., Fischer, 1976; Prandle, 1981; Thatcher and Najarian, 1983).

Interestingly, using equations (2) and (4) we can derive the dispersion as a function of the salinity gradient (Savenije, 2015):

$$\frac{D}{D_1} = \left(-\frac{AD_1}{|Q_f|s_1}\frac{\mathrm{d}s}{\mathrm{d}x}\right)^{\frac{K}{1-K}} \tag{5}$$

which connects the dispersion coefficient to local variables ($A$, $\mathrm{d}s/\mathrm{d}x$), boundary conditions ($D_1$, $s_1$), and $K$.

MacCready (2004, 2007) derived an equation for the exchange term theoretically:

$$\overline{u's'} - K_H\frac{\mathrm{d}s}{\mathrm{d}x} = \left(m_1\frac{h^2 u_f{}^2}{K_S} + K_H\right)\left(-\frac{\mathrm{d}s}{\mathrm{d}x}\right) + m_2\frac{gc_s h^5 u_f}{K_S K_E}\left(-\frac{\mathrm{d}s}{\mathrm{d}x}\right)^2 + m_3\frac{g^2 c_s{}^2 h^8}{K_S K_E{}^2}\left(-\frac{\mathrm{d}s}{\mathrm{d}x}\right)^3 = -D\frac{\mathrm{d}s}{\mathrm{d}x} \tag{6}$$

where $\overline{u's'}$ is the tidal average exchange flow salt flux, $u'$ is the depth varying velocity, $s'$ is the depth varying salinity, $K_H$ is the along channel diffusion coefficient, $m_1 = \frac{2}{105}$, $m_2 = \frac{19}{420 \times 48}$ and $m_3 = \frac{19}{630 \times 48^2}$ are constant values following MacCready's





vertical integration, $u_f = |Q_f|/A$ is the depth-averaged velocity of fresh water, $K_S$ is the effective vertical eddy diffusivity, $g$ is the gravity acceleration, $c_s$ is the saline expansivity equal to $7.7 \times 10^{-4}$, and $K_E$ is the effective hydraulic eddy viscosity. For the latter, we use the equation $K_E = 0.1 \frac{2}{\pi} u_* h$, with $u_* = \frac{\sqrt{g}}{C} \upsilon$ as the shear velocity in relation to the tidal velocity amplitude $\upsilon$, where $C = K_m h^{1/6}$ is the coefficient of Chézy, and $K_m$ is Manning's coefficient. Comparing the salt balance equation of Maccready to equation (2) implies that equation (6) is identical to $-D \frac{ds}{dx}$. MacCready assumed the estuary to be narrow and rectangular, in the sense that cross-sectional shape does not basically modify the width-averaged dynamics. In the derivation, he also assumed the effective vertical eddy viscosity to be constant with depth, after Hansen and Rattray (1965), and that the salinity gradient of the depth-varying part is much smaller than the depth-averaged part, after Pritchard (1952). Additionally, other effects like salt storage, internal hydraulics and the Coriolis force were considered negligible.

After division of all terms by the salinity gradient, it becomes an equation for the dispersion coefficient $D$:

$$D = \left( m_1 \frac{h^2 u_f{}^2}{K_S} + K_H \right) + m_2 \frac{g c_s h^5 u_f}{K_S K_E} \left( -\frac{ds}{dx} \right) + m_3 \frac{g^2 c_s{}^2 h^8}{K_S K_E{}^2} \left( -\frac{ds}{dx} \right)^2 \tag{7}$$

whereby the first term is not dependent on the salinity gradient, the second is directly proportional to it, and the third term depends on the square of the salinity gradient.

Based on equation (5) we can also derive an expression for the dispersion:

$$D = D_1 \left( \frac{A_1 D_1}{l |Q_f|} \right)^{\frac{K}{1-K}} \left( -\frac{A}{A_1} \frac{l}{s_1} \frac{ds}{dx} \right)^{\frac{K}{1-K}} \tag{8}$$

where $A_1$ is the cross-sectional area at the inflection point (at $x = x_1$), $l = L - x_1$ is the distance from the inflection point to where salinity becomes the same as the fresh water salinity, $L$ is the local intrusion length.

Hence $D \propto \gamma^{\left( \frac{K}{1-K} \right)}$ with $\gamma = -\frac{A}{A_1} \frac{l}{s_1} \frac{ds}{dx}$. Given the function $F(\gamma) = \gamma^{\left( \frac{K}{1-K} \right)}$, a Taylor series expansion near $\gamma = 1$ can be derived as:

$$F(\gamma) = \frac{(2K-1)(3K-2)}{2(1-K)^2} + \frac{K(2-3K)}{(1-K)^2} \left( \frac{A}{A_1} \frac{l}{s_1} \right) \left( -\frac{ds}{dx} \right) + \frac{K(2K-1)}{2(1-K)^2} \left( \frac{A}{A_1} \frac{l}{s_1} \right)^2 \left( -\frac{ds}{dx} \right)^2 + R_2(x) \tag{9}$$

where $R_2(x)$ is the residual term, considered to be small. To analysis the importance of the different terms of (9), Figure 1 presents the factors $g_1 = \frac{(2K-1)(3K-2)}{2(1-K)^2}$, $g_2 = \frac{K(2-3K)}{(1-K)^2}$ and $g_3 = \frac{K(2K-1)}{2(1-K)^2}$. $g_1$ is the closure term which compensates for $g_2$ and $g_3$ so as to make $\sum g_i = 1$ $(i = 1, 2, 3)$. It is clear that the absolute value of the first term is much smaller than the density-driven terms. Also, the larger the value of $K$, the more important the third term is. This is in accordance with traditional literature if we analyze equation (9) mathematically. When $K = 1/2$, $F(\gamma) = \left( \frac{A}{A_1} \frac{l}{s_1} \right) \left( -\frac{ds}{dx} \right)$, dispersion is proportional to the salinity gradient, meaning it driven by the longitudinal salinity gradient. If $K = 2/3$, $F(\gamma) = \left( \frac{A}{A_1} \frac{l}{s_1} \right)^2 \left( -\frac{ds}{dx} \right)^2$ and dispersion is proportional to the square of the salinity gradient, which according to Fischer (1976) means that the dispersion is dominated by lateral exchange. According to these theoretical studies, $K$ is in a range of 1/2 to 2/3, which is consistent with the situation where the dispersion combines longitudinal and lateral effects. The larger the value of $K$, the stronger the effect of lateral dispersion.

Considering only the density-driven terms in equations (7) and (9), the proportionality results in:

$$\frac{2-3K}{2K-1} = 36 \frac{K_E |Q_f|}{g c_s h^3 A_1} \frac{l}{s_1} = \frac{7.2 E |Q_f|}{\sqrt{g} c_s h^2 A_1 C T} \frac{l}{s_1} = w \tag{10}$$





leading to an analytical expression for $K$:

$$K = \frac{2+w}{3+2w} \tag{11}$$

Since $w > 0$, the upper limit value of $K$ is $2/3$ if the river discharge is small or the cross section is large and deep and $w$ approaches zero. If the river discharge is large or the cross section is small and shallow, $w$ is large and $K$ converges to a value of $1/2$. We have used this expression to compute $K$ values in real estuaries using the database of Savenije (2012). These $K$ values are in a range of 0.51-0.64 (see Section 4.3).

Overall, there are three results for the estimation of Van der Burgh's coefficient: 1) from comparison with theoretical studies ($K = 1/2$ or $K = 2/3$), 2) by comparison between the salinity driven mixing terms ($1/2 < K < 2/3$), and 3) based on the empirical data set obtained by equation (11). These results are surprisingly close. In general, it can be concluded that the comparison between the theoretical work of MacCready and the analytical equations based on the Van der Burgh's relationship provides a solid foundation for the use of Van der Burgh's equation in one-dimensional analytical models.

## 3 Including Residual Circulation in Wide Estuaries

In the theory about mixing in estuaries, several authors have distinguished between tide-driven and density-driven dispersion (e.g., Hansen and Rattray, 1965; Banas et al., 2004). The tide is an active hydraulic driver that creates shear stresses in the flow as momentum, resulting from friction along the boundaries, transferred to the heart of the channel by turbulence. Generally these shear stresses reduce stratification and hence reduce dispersion. However, at large scale, the tide facilitates mixing by tidal trapping and residual circulation, which enhances dispersion. Tidal trapping results from irregularities of the channel, leading to pockets of relatively high or low salt concentrations that later reunite with the stream. The mixing length scale of tidal trapping is the tidal excursion. By using the tidal excursion as the mixing length, tidal trapping can be incorporated in a predictive equation. Residual circulation is more complicated. It can be a very powerful tide-driven mechanism in the wider parts of estuaries where the tide causes mixing by the cross-over of preferential ebb and flood channels that develop in wide estuaries, such as the Schelde, described by Nguyen et al. (2008). But how can we parameterize residual circulation? Here a different approach is followed from Nguyen et al. (2008), trying to combine this effect in the regular one-dimensional advection-dispersion equation.

### 3.1 Model Including Residual Circulation

Figure 2 presents the sketch of a box model used to include lateral exchange into longitudinal dispersion. Water particles in the middle can mix longitudinally and laterally within their respective mixing lengths. For the longitudinal mixing length we consider the tidal excursion and for the lateral exchange the half of estuary width. The balance of mass can then be described as:

$$\frac{V \Delta s_2}{\Delta t} = |Q_f|(s_2 - s_1) + d(s_1 - 2s_2 + s_3) + r(s_L - 2s_2 + s_R) \tag{12}$$





where $V = AE$ is the water volume, $E$ is tidal excursion length, $s$ is salinity, with subscripts of different locations, $d$ and $r$ are longitudinal and lateral exchange flows.

The balance equation then becomes:

$$V\frac{\partial s}{\partial t} - |Q_f|\frac{\partial s}{\partial x}\Delta x - d\frac{\partial^2 s}{\partial x^2}(\Delta x)^2 - r\frac{\partial^2 s}{\partial y^2}(\Delta y)^2 = 0 \tag{13}$$

where $\Delta x$ and $\Delta y$ are the mixing lengths, which are taken as $\Delta x = E$ and $\Delta y = B/2$.

The assumption used is that the lateral exchange is proportional to longitudinal (Fischer, 1972),

$$r\frac{\partial^2 s}{\partial y^2} \propto d\frac{\partial^2 s}{\partial x^2} \tag{14}$$

As a result, longitudinal and lateral processes can be combined into one single one-dimensional equation:

$$\frac{\partial s}{\partial t} - \frac{|Q_f|}{A}\frac{\partial s}{\partial x} - \frac{dE}{A}\left(1 + C_2\left(\frac{B}{E}\right)^2\right)\frac{\partial^2 s}{\partial x^2} = 0 \tag{15}$$

Comparing (15) with the traditional salt balance equation, the effective longitudinal dispersion is:

$$D = \frac{dE}{A}\left(1 + C_2\left(\frac{B}{E}\right)^2\right) \tag{16}$$

Subsequently, the longitudinal exchange flow $d$ is assumed to be proportional to the amplitude of the tidal flow (driving the circulation) $(\widehat{Q_t} = Av)$, and to the stratification number to the power of Van der Burgh's coefficient:

$$d = C_1(N_R)^K\widehat{Q_t} \tag{17}$$

with $N_R$ defined as the ratio of potential energy of the river discharge to the kinetic energy of the tide over a tidal period:

$$N_R = \frac{\Delta\rho}{\rho}\frac{gh}{v^2}\frac{|Q_f|T}{AE} \tag{18}$$

where $\Delta\rho/\rho = c_s s$ is the relative density difference between river water and saline water and $T$ is tidal period.

The reason why the exchange flow is a function of the stratification number to the power of $K$ is because it is in agreement with equation (4), $\Delta\rho/\rho$ being directly proportional to $s$.

We then obtain a simple dimensionless expression for the dispersion coefficient, simulating to the one by Gisen et al. (2015b) but incorporating lateral exchange flow:

$$\frac{D}{vE} = C_1(N_R)^K\left(1 + C_2\left(\frac{B}{E}\right)^2\right) \tag{19}$$

where $C_1$ and $C_2$ are constants .

### 3.2 Analytical Solution

In almost all estuaries, the ratio of width to excursion length is quite small, particularly upstream where salinity intrusion happens. So for further analytical solutions we can focus on the first part of equation (19):

$$D = C_1(N_R)^K vE \tag{20}$$




The traditional approach by Savenije (2012) merely uses this equation as the boundary condition at $x = x_1$, after which $D(x)$ values are obtained by integration of the Van der Burgh's equation along the estuary axis. But, in principle, with this equation the dispersion can be calculated at any point along the estuary, provided local hydraulic and geometric variables are known. Using $\upsilon = \pi E/T$, equation (20) then becomes:

$$D(x) = C_1 (c_s g \pi)^K \left( \frac{s|Q_f|}{\upsilon^3 B} \right)^K \upsilon E \tag{21}$$

5  where all local variables are now function of $x$.

The following equations are used for the tidal velocity amplitude, width and tidal excursion:

$$\upsilon(x) = \upsilon_1 e^{\delta_\upsilon (x - x_1)} \tag{22}$$

$$B(x) = B_1 e^{\left( -\frac{x - x_1}{b} \right)} \tag{23}$$

$$E(x) = E_1 e^{\delta_H (x - x_1)} \tag{24}$$

where $\delta_\upsilon \approx \delta_H$ are damping/amplifying rate of tidal velocity amplitude and tidal range, $b$ is the width convergence length ($b_1$ downstream of the inflection point and $b_2$ upstream).

At the inflection point, the predicted equation is given by:

$$D_1 = C_1 (c_s g \pi)^K \left( \frac{s_1 |Q_f|}{\upsilon_1^3 B_1} \right)^K \upsilon_1 E_1 \tag{25}$$

where the subscript '1' means parameters are evaluated at the inflection point ($x = x_1$).

Substitution of equations (22)-(25) in (21) gives:

$$D(x) = D_1 \left( \frac{s}{s_1} \right)^K e^{\Omega (x - x_1)} \tag{26}$$

with $\Omega = 2\delta_H - 3K\delta_H + K/b$.

Differentiating $D$ with respect to $x$ and using equation (26) results in:

$$\frac{\mathrm{d}D}{\mathrm{d}x} = K \frac{D}{s} \frac{\mathrm{d}s}{\mathrm{d}x} + \Omega D \tag{27}$$

Combining the result with the time-averaged salt balance, equation (27) results in:

$$\frac{\mathrm{d}D}{\mathrm{d}x} = \Omega D - K \frac{|Q_f|}{A} \tag{28}$$

For a prismatic channel ($b \to \infty$) with constant width and little tidal damping, $\Omega = 0$ and (28) becomes the Van der Burgh's

equation. As a result, the exponent of $N_R$ in this model represents the Van der Burgh's coefficient.




The cross-sectional area $A$ is given by:

$$A(x) = A_1 e^{\left(-\frac{x-x_1}{a}\right)} \tag{29}$$

where $a$ is the cross-sectional convergence length ($a_1$ downstream of the inflection point and $a_2$ upstream).

Substitution of equation (29) in (28) gives:

$$\frac{\mathrm{d}D}{\mathrm{d}x} = \Omega D - K\frac{|Q_f|}{A_1} e^{\left(\frac{x-x_1}{a}\right)} \tag{30}$$

In analogy with Kuijper and Van Rijn (2011), the solution of this linear differential equation is:

$$\frac{D}{D_1} = \left[ e^{\Omega(x-x_1)} + \frac{K|Q_f|}{A_1 D_1}\zeta\left(e^{\Omega(x-x_1)} - e^{(x-x_1)/a}\right)\right] \tag{31}$$

with $\zeta = \frac{a}{1-\Omega a}$.

The maximum salinity intrusion length is obtained from equation (31) after substitution of $D \to 0$ at $x = L$:

$$L = \zeta \ln\left(\frac{A_1 D_1}{K|Q_f|\zeta} + 1\right) + x_1 \tag{32}$$

This is the same equation as in Savenije (2005) if $\zeta = a$.

Using equation (26), the longitudinal salt distribution becomes:

$$\frac{s}{s_1} = \left[1 + \frac{K|Q_f|}{A_1 D_1}\zeta\left(1 - e^{(x-x_1)/\zeta}\right)\right]^{1/K} \tag{33}$$

This solution is similar to the solution by Kuijper and Van Rijn (2011), with the difference that Kuijper and Van Rijn used a constant value of $K = 0.5$ and that their value of $\Omega$ depended on bottom slope.

So with these new analytical equations, the local dispersion and salinity can be obtained, using boundary condition at the inflection point. This method is limited since it only works when $B/E < 1$. If we want to account for residual circulation using equation (19), then we have to use numerical integration of equation (2) using (19) for $D$.

## 4 Empirical Validation

### 4.1 Summary Information

18 estuaries with quite different characteristics, covering a diversity of sizes, shapes and locations, have been selected from the database of Savenije (2012). It appears that all these alluvial estuaries can be schematized in one or two segments separated by a well-defined inflection point (Savenije, 2015). As an example, Figure 3 shows the geometry of two estuaries: Maputo with inflection point and Thames without inflection point. It can be seen that the natural geometry fits well on semilogarithmic paper, indicting an exponential variation of the cross section and width. Geometric data of all 18 estuaries are presented in Appendix A.

In Table 1 the general geometry of estuaries are summarized, where $B_f$ is the bankfull stream width. It is obvious that these estuaries cover a wide range of sizes. Estuary with $x_1 = 0$ means there is no inflection point. In addition, the larger the $a_2$



($b_2$), the slower the cross section (width) declines upstream. With a large $b_2$, a relatively small value of $b_1$ suggests the channel with a pronounced funnel shape with fast decrease of width near the mouth. In contrast, a relatively large value of $b_2$ indicates estuaries with near-prismatic shape. The same values of $a$ and $b$ indicate that the depth is constant.

Table 2 and 3 contain summary information of estuaries on different measurement dates, where $H_1$ is the tidal range at $x_1$, $\eta$ is tidal amplitude, $\alpha = \frac{D_1}{|Q_f|}$ is the mixing coefficient and $\beta = \frac{Ka_2|Q_f|}{A_1D_1}$ is the dispersion reduction ratio. Tidal excursion and tidal period are more or less the same in all estuaries, except Lalang and Chao Phraya with a diurnal tide. Most estuaries damp upstream, with negative values of $\delta_H$. In addition, most estuaries have a small tidal amplitude to depth ratio, which means relatively simple solutions of hydraulic equations are possible (Savenije, 2005). $K$ values have been obtained by calibration of simulated salinities to observations in 18 estuaries. Using an automatic solver, the best result was obtained with $C_1 = 0.10$, $C_2 = 12$ and $K = 0.58$. For individual estuary, $K$ values were obtained ranging between 0.45-0.78. In some estuaries $K > 2/3$, which is because the empirical values only have the mathematically feasible limitation of $0 < K < 1$, and are not subject to the assumptions made in the theoretical derivation as by MacCready (2004). The dispersion at the inflection point has a range of 50-600 $\mathrm{m}^2/\mathrm{s}$ in a diversity of estuaries which is consistent with Prandle (1981). The mixing coefficient demonstrates to what extent the dispersion overcomes the flushing by river flow. The larger the river discharge, the smaller the $\alpha$, meaning it is difficult for the salinity to penetrate into the estuary. The dispersion reduction ratio determines the longitudinal variation of dispersion. Fischer et al. (1979) suggested that the transition from a well-mixed to a strongly stratified estuary occurs when the values of stratification number $N_R$ are in the range of 0.08-0.8. With a ratio of $\pi$ between Fischer's and our expressions for the stratification number, the range becomes 0.25-2.51. It is obvious that all estuaries are partially- to well-mixed with $N_R$ below 2.51.

## 4.2 Sensitivity to $C_2$

Through the use of $C_2$ we can use a single dispersion equation accounting for two-dimensional effects in a one-dimensional model. The assumption that lateral exchange is proportional to longitudinal dispersion suggests $C_2$ to be independent of $x$. Figure 4 and Appendix B demonstrate how salinity changes with varying $C_2$. Salinities were simulated by numerical solution of equation (2) with (19) based on the boundary condition at $x = x_1$. Typically, $C_2$ matters mainly near the mouth, but there is almost no effect on narrow estuaries like Lalang, Limpopo, Tha Chin and Chao Phraya. Hence, the inclusion of the residual circulation improves the accuracy of salinity simulation in wide estuaries and more particularly near the mouth of the estuaries where the ratio of width to tidal excursion is relatively large.

To check the sensitivity to $C_2$, values of 1, 10 and 50 have been used to calculate salinity curves. It is demonstrated that the larger the value of $C_2$, the smaller the salinity gradient and the flatter the salinity curve near the estuary mouth. However, because of the interdependence of $D$, $s$ and $\mathrm{d}s/\mathrm{d}x$ through equation (2) in the upstream part, a larger value of $C_2$ can lead to larger salinities (e.g., Thames, Elbe, Edisto, Maputo and Corantijn). Basically, $C_2 = 10$ (green lines) can perform perfectly in 14 out of 18 estuaries (e.g., Maputo and Thames). We can see that larger values than $C_2 = 10$ cause exaggerated salinity in the downstream part of these estuaries which is why a general values of $C_2 = 10$ is recommended. The poorer results occur in estuaries that have peculiar shapes near the mouth. A larger value of $C_2$ applies to the Kurau. This may be because the





width is underestimated in the wide estuary mouth, due to misinterpretation of the direction of the streamline (The width is determined according to a line perpendicular to the streamline). As a matter of fact, the width should be larger and dispersion should be larger with smaller salinity gradients, which would then result in a lower value of $C_2$. The same applies to Endau. On the contrary, A smaller value of $C_2$ in Perak fits better, because of overestimating of the width. Here the topographical map suggests a wider estuary mouth, whereas the tidal flow is concentrated in a much narrower main channel due to a the north

bank protruding into the estuary and a spur from the south projecting into the mouth. The Selangor has a similar situation. It shows that the configuration of the mouth is important for the correct simulation of the salinity near the estuary mouth. But, fortunately, a relatively poor performance near the mouth of these estuaries does not affect the salinity distribution upstream as long as $C_2$ is not too large. In conclusion, $C_2 = 10$ appears to be a suitable default value as long as the trajectory of the tidal currents can be considered properly.

The poor fit in the downstream parts of the Lalang and Chao Phraya, in which measured salinities are lower than simulated, can be explained by a complex downstream boundary. The Lalang estuary has a pronounced riverine character and is a tributary to the complex estuary system of the Banyuasin, sharing its outfall with the large Musi river. So the salinity near its mouth is largely affected by the Musi. Also, pockets of fresh water can decrease the salinity near the confluence. The Chao Phraya opens to the Gulf of Thailand where the salinity is influenced by historical discharges rather than ocean salinity, remaining relatively

fresh. Other measurement uncertainties may cause outliers as well.

### 4.3  What Determines $K$

The physical meaning of the Van der Burgh's coefficient has been analyzed linking it to traditional theoretical research. Equation (10) shows a direct relation between this coefficient and MacCready's parameters which are measurable quantities. Hence, the coefficient is affected by tide, geometry and fresh water discharge. Gisen (2015a) assumed $K$ to be independent on $Q_f$

and obtained a power function for predicting $K$, in which all parameters merely depended on the topography. Shaha and Cho (2011) also found $K$ values to depend on river discharge and considered its value to increase upstream in a range of 0-1 due to different mechanisms along the estuary. Yet, from equation (10), the $K$ value should be constant if the depth is constant along the estuary.

A 1:1 plot is presented in Figure 5, relating the empirical $K$ values to the predicted values using equation (11). The predicted

$K$ values have a smaller range (0.51-0.64) than the calibrated ones (0.45-0.78). However, they are quite similar considering they have been obtained from different approaches. This correspondence strengthens the physical basis of the Van der Burgh's equation. All $K$ values are very close to $0.58$ which may be a good starting value in estuaries where information on geometry and channel roughness is lacking.

### 5  Discussion and Conclusion

Overall, the single one-dimensional salinity intrusion model including residual circulation appears to work well in natural estuaries with a diversity of geometric and tidal characteristics, both by analytical and numerical computation. The new equation




is a simple and useful tool for analyzing local dispersion and salinity directly on the basis of local hydraulic variables. In a calibration mode, $K$ is the only parameter to be calibrated using $C_1 = 0.10$ and $C_2 = 10$. In a predictive mode, a value of $K = 0.58$ can be used as a first estimate. If information on river discharge, roughness and geometry is available, $K$ can be determined iteratively by taking $K = 0.58$ as the predictor and subsequently substituting $s_1$ and $l$ from the first iteration by equations (10) and (11) and repeating the procedure until the process converges.

5  The addition of the factor $(1 + C_2(B/E)^2)$ in the dispersion equation proved valuable near the mouth of estuaries where residual circulation due to interacting ebb and flood channels dominate dispersion. The value $C_2 = 10$ was found to perform best in most estuaries, indicating that residual circulation is dominant in wide estuaries where ebb and flood currents prevail.

  Van der Burgh's coefficient determines the way dispersion relates to the stratification number by a power function. Two approaches, theoretical derivation from traditional literature and empirical validation based on observations in a large set of 10 estuaries, provided similar estimates of Van der Burgh's coefficient. Under MacCready's assumptions, there are three ways to estimate $K$: $0.51 < K < 0.64$ from empirical application of equations (10) and (11); $1/2 < K < 2/3$ as the physical boundaries of (11); and the comparison with traditional approximations ($K = 1/2$ or $K = 2/3$). After calibration of the new analytical model to the database of field observations, the values of $K$ were in a range of 0.45-0.78 for a wide range of conditions, with an average of 0.58, close to the predicted values. MacCready's equation determines dispersion via a decomposition method, using 15 depth-varying velocity and salinity. Although these 1-D expressions of velocity and salinity may be simplifications of reality, the good correspondence between Van der Burgh's equation and MacCready's theory provides a strong theoretical basis for Van der Burgh's equation.

  A previous analytical salinity intrusion model was developed by Gisen (2015a), from which the $K$ values resulted in a range of 0.20-0.75 by calibration and 0.22-0.71 by prediction. These solutions cover a wider range than our estimates because of 20 Gisen's assumption that $K$ does not depend on river discharge and because of three improvements made in this paper. Firstly, we used the local hydraulic parameters to simulate the salinities, while Gisen used a constant depth and no damping ($\Omega = 0$). In addition, by using an uncertainty bound of 25 % on fresh water discharge we could reduce the inaccuracy of the tail of the salinity curve and obtain a better fit (where $K$ matters most). And finally, all geometric analyses were improved by revisiting the fit to observations.

25  An important consequence of this research is that $K$ depends on the water discharge. Where Gisen assumed $K$ to be constant for each estuary, we find substantial variability for estuaries where a larger range of discharges is available: e.g. in the Maputo $0.57 < K < 0.70$; in the Limpopo $0.61 < K < 0.72$; and in the Edisto $0.48 < K < 0.58$. The implication of discharge dependence needs to be tested further for predictive purpose.

  In some particular cases, the simulated salinity with $C_2 = 10$ does not fit the observations near the estuary mouth. So one 30 should be aware of peculiar configurations of streamlines and geometries near the estuary mouth when using this model. MacCready and Geyer (2010) also pointed out that the effect of irregular channel shape is important. However, a poor fit near the estuary mouth has almost no effect on the total salinity intrusion length. It is suggested that in future research the assumption that lateral exchange is proportional to longitudinal exchange needs to be tested further. Finally, this predictive one-dimensional salinity intrusion model, having a stronger theoretical basis, may be a useful tool in ungauged estuaries.





## Appendix A: Notation

| Symbol | Meaning | Dimension | Symbol | Meaning | Dimension |
|--------|---------|-----------|--------|---------|-----------|
| $a$ | cross-sectional convergence length | [L] | $Q_t$ | amplitude of tidal flow | [L$^3$/T] |
| $A$ | cross-sectional area | [L$^2$] | $r$ | lateral exchange flow | [L$^3$/T] |
| $b$ | width convergence length | [L] | $R_2(x)$ | residual term | [-] |
| $B$ | width | [L] | $s$ | salinity | [M/L$^3$] |
| $B_f$ | bankfull stream width | [L] | $s_f$ | fresh water salinity | [M/L$^3$] |
| $C$ | coefficient of Chézy | [L$^{1/2}$/T] | $s'$ | depth varying salinity | [M/L$^3$] |
| $C_i$ | constant | [-] | $t$ | time | [T] |
| $c_s$ | saline expansivety | [-] | $T$ | tidal period | [T] |
| $d$ | longitudinal exchange flow | [L$^3$/T] | $u$ | flow velocity | [L/T] |
| $D$ | dispersion coefficient | [L$^2$/T] | $u'$ | depth varying flow velocity | [L/T] |
| $E$ | tidal excursion length | [L] | $u_f$ | velocity of fresh water | [L/T] |
| $g$ | gravity acceleration | [L/T$^2$] | $u_*$ | shear velocity | [L/T] |
| $g_i$ | factor | [-] | $V$ | water volume | [L$^3$] |
| $h$ | depth | [L] | $x$ | distance | [L] |
| $H$ | tidal range | [L] | $\alpha$ | mixing coefficient | [L$^{-1}$] |
| $K$ | Van der Burgh's coefficient | [-] | $\beta$ | dispersion reduction ratio | [-] |
| $K_H$ | diffusion coefficient | [L$^2$/T] | $\gamma$ | dimensionless argument | [-] |
| $K_E$ | hydraulic eddy viscosity | [L$^2$/T] | $\delta$ | damping/amplifying rate | [L$^{-1}$] |
| $K_m$ | Manning's coefficient | [L$^{1/3}$/T] | $\Delta x, \Delta y$ | mixing lengths | [L] |
| $K_S$ | vertical eddy diffusivity | [L$^2$/T] | $\upsilon$ | tidal velocity amplitude | [L/T] |
| $l$ | intrusion length from inflection point | [L] | $\zeta$ | adjusted convergence length | [L] |
| $L$ | intrusion length | [L] | $\eta$ | tidal amplitude | [L] |
| $m_i$ | constant | [-] | $\rho$ | density of water | [ML$^{-3}$] |
| $N_R$ | stratification number | [-] | $\Omega$ | adjustment parameter | [L$^{-1}$] |
| $Q_f$ | fresh water discharge | [L$^3$/T] | | | |



## Appendix B:  Compilation of the geometry

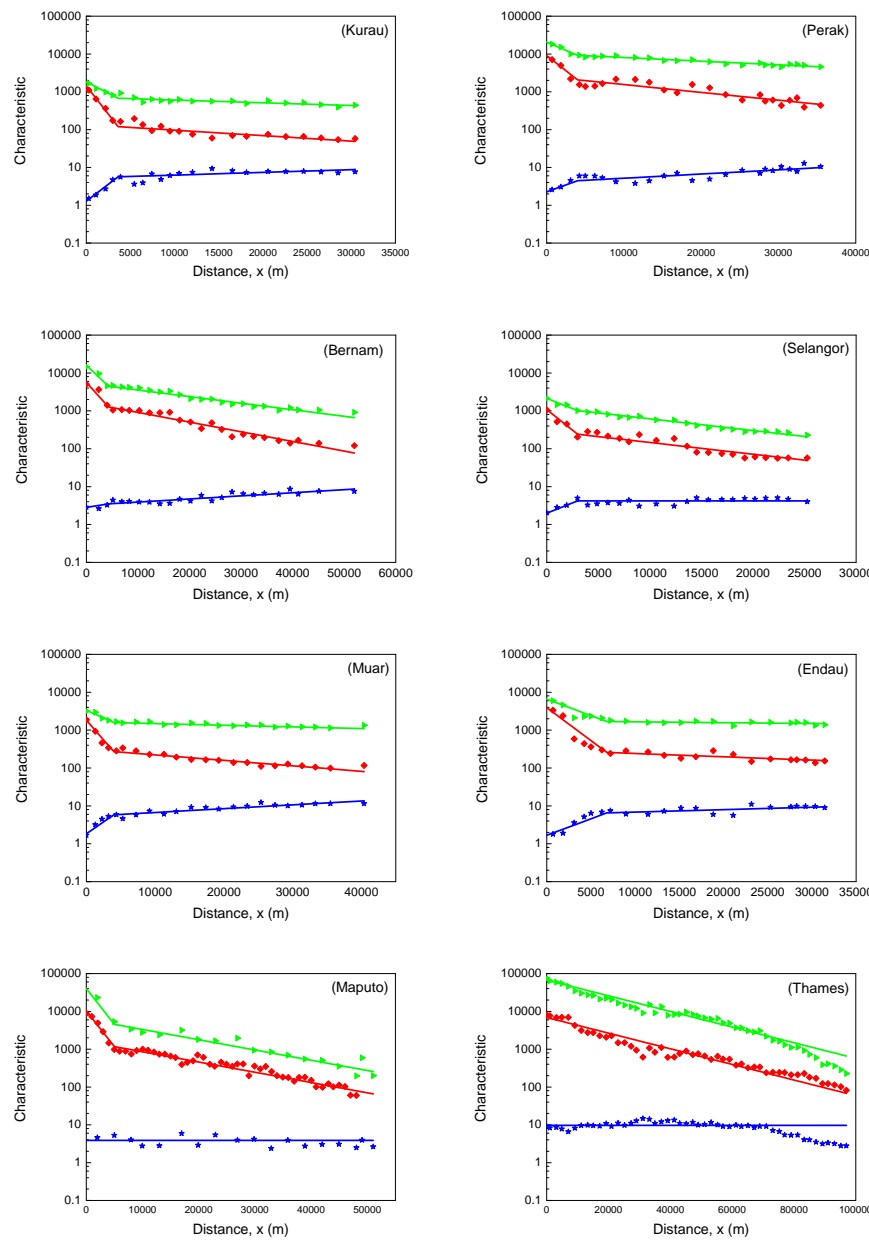





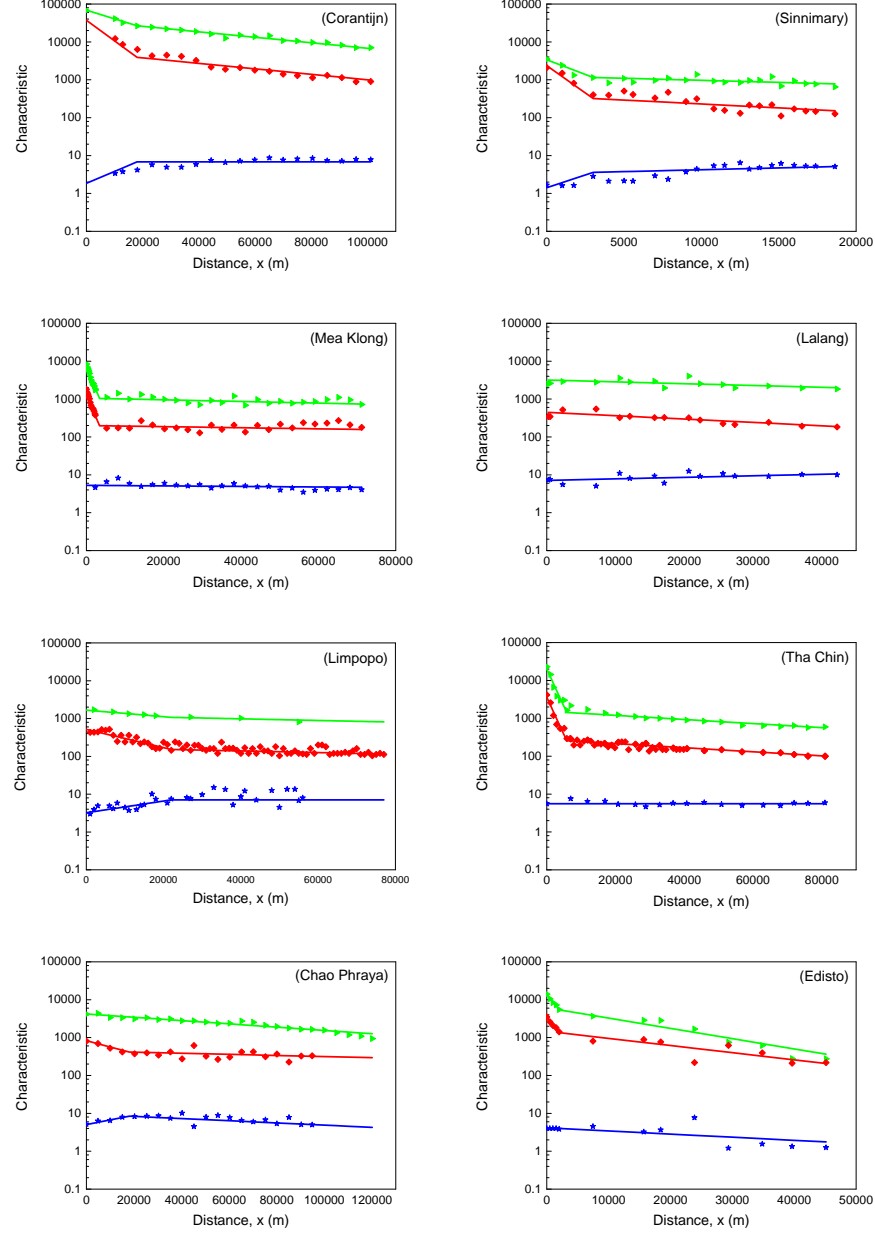

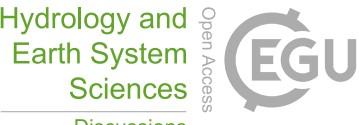



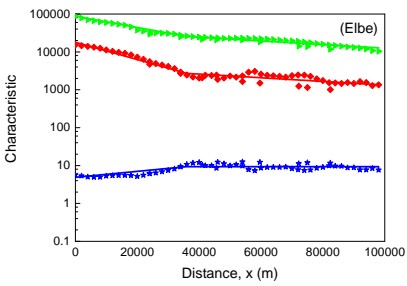 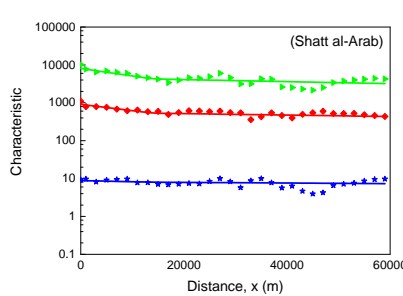

## Appendix C: Sensitivity to $C_2$

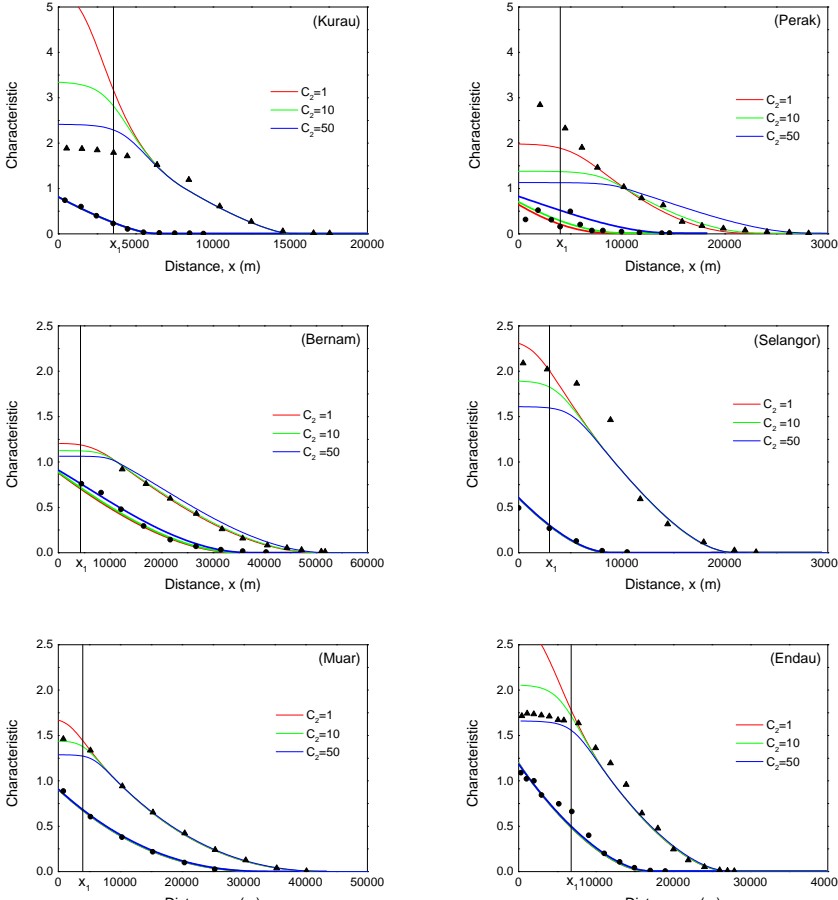





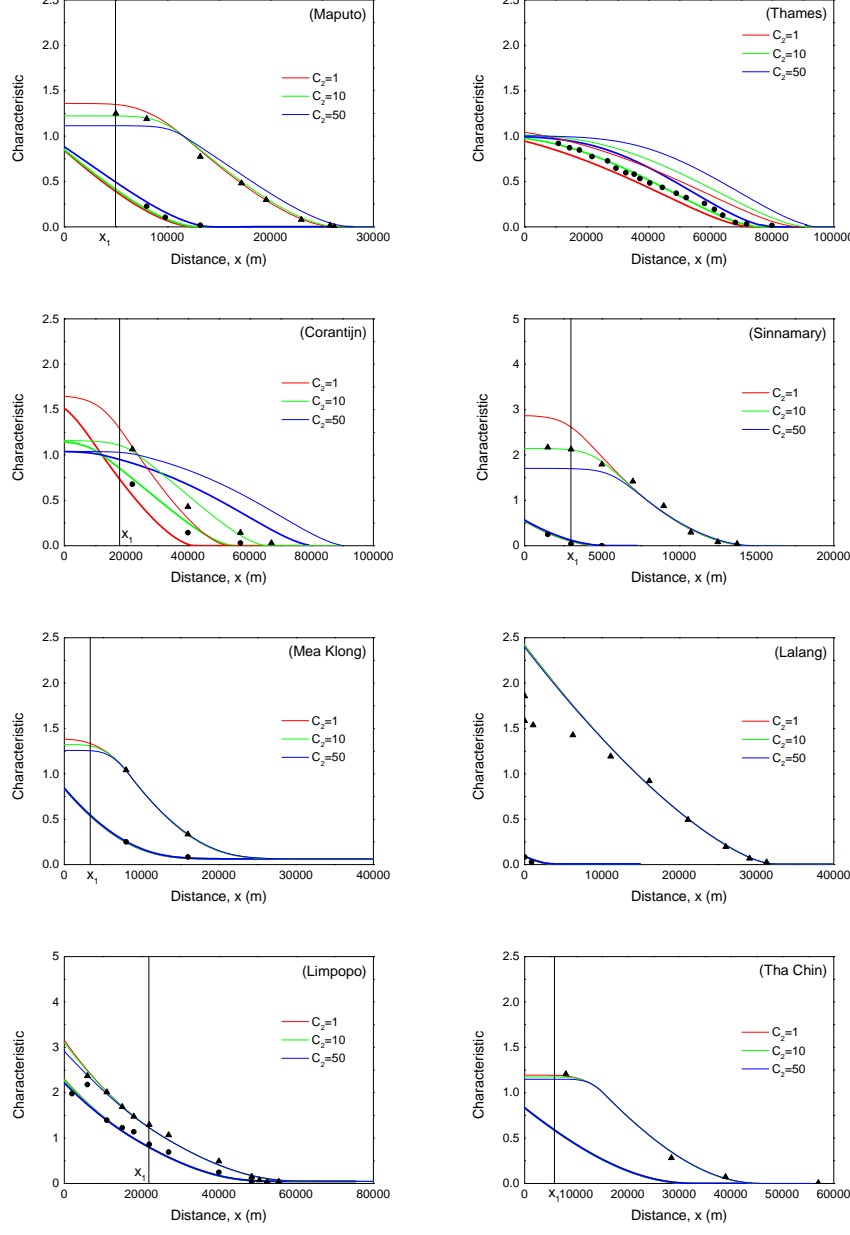

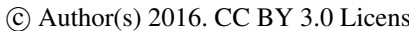





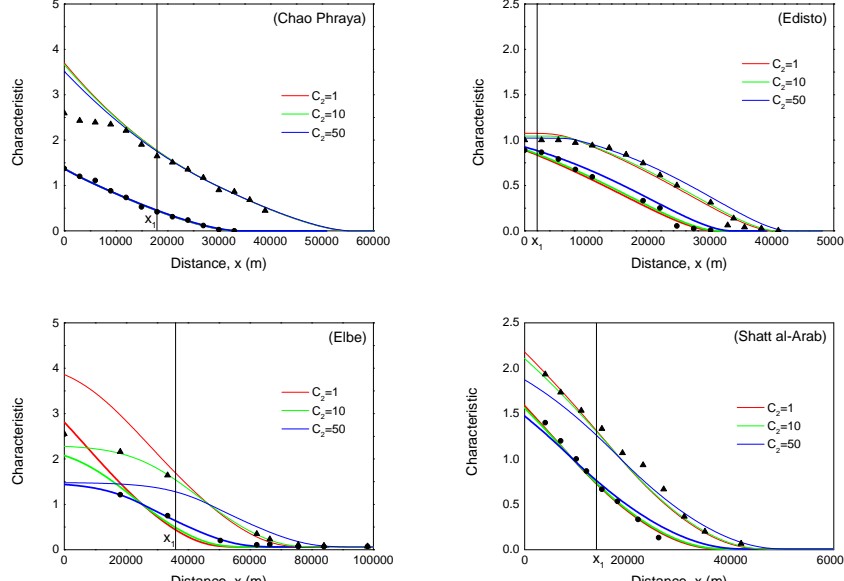

*Acknowledgements.* The first author is financially supported for her Ph.D. research by the China Scholarship Council.





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




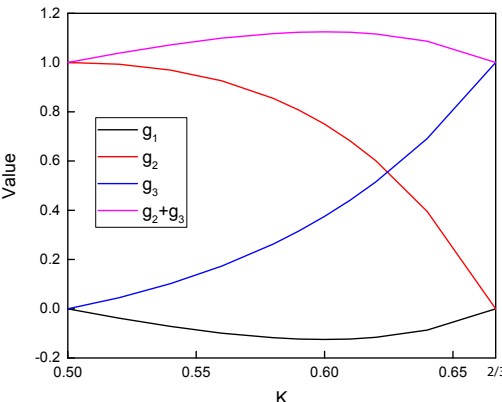

**Figure 1.** Comparison between the factors in the Taylor series expansion of $F(\gamma)$ as a function of the Van der Burgh coefficient $K$.

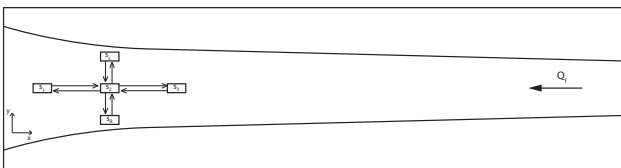

**Figure 2.** Conceptual sketch for lateral and longitudinal mixing. Longitudinal and lateral mixing lengths are $\Delta x$ and $\Delta y$, respectively.

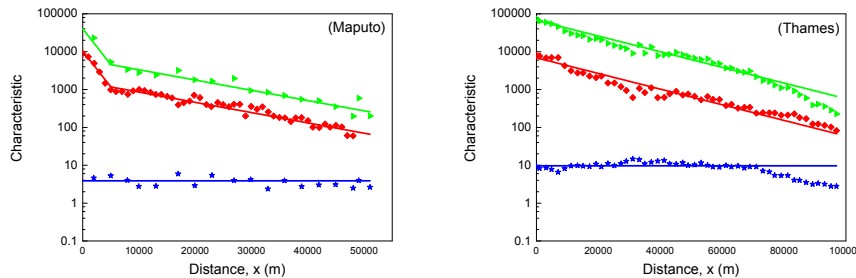

**Figure 3.** Semi-logarithmic presentation of estuary geometry, comparing simulated (lines) to the observations (symbols), including cross-sectional area (green), width (red) and depth (blue).





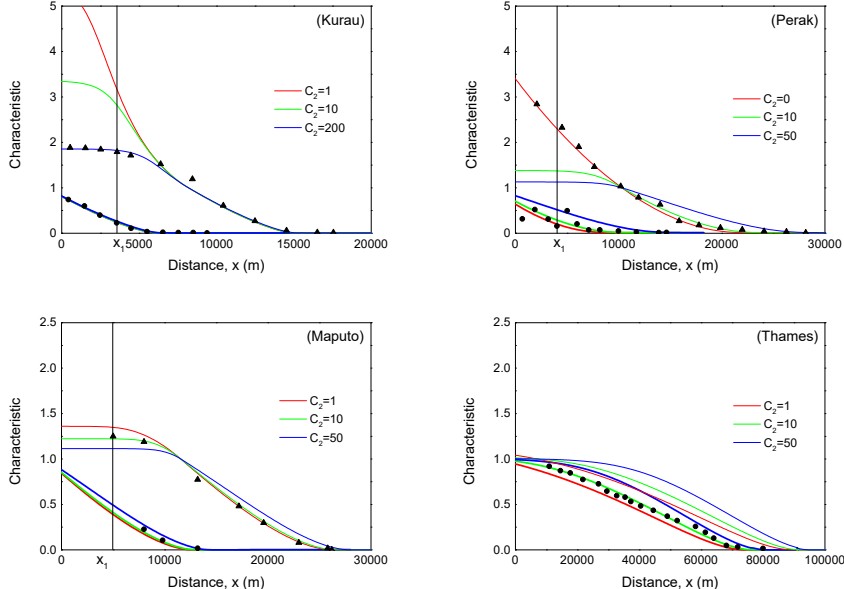

**Figure 4.** Comparison between simulated and observed salinity at high water slack (thin lines) and low water slack (thick lines), scaled by the salinity $s_1$ at the inflection point $x_1$ for different $C_2$ values. Observations at high water slack are represented by triangles and low water slack by circles. Observe that the Thames only has low water slack observations.

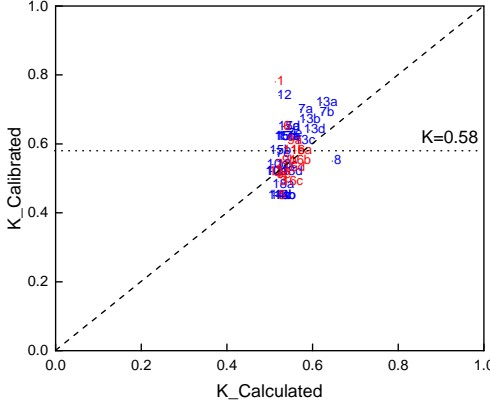

**Figure 5.** Comparison between predicted and calibrated $K$ values. Labels are used to distinguish estuaries. The blue marks used $K_m$ from Cai et al. (2012) and the red ones from Gisen (2015a).





**Table 1.** Summary of the geometry of the estuaries

| Label | Estuary | $A_1$ [m²] | $a_1$ [km] | $a_2$ [km] | $b_1$ [km] | $b_2$ [km] | $x_1$ [m] | $h_1$ [m] | $B_f$ [m] | $K_m$ [m$^{1/3}$s$^{-1}$] |
|---|---|---|---|---|---|---|---|---|---|---|
| 1 | Kurau | 674 | 3.6 | 60 | 1.45 | 30 | 3600 | 5.6 | 20 | 30 |
| 2 | Perak | 9212 | 5 | 45 | 2.7 | 21 | 4000 | 4.5 | 130 | 65 |
| 3 | Bernam | 4460 | 3.4 | 25 | 2.9 | 17 | 4300 | 3.5 | 45 | 70 |
| 4 | Selangor | 1015 | 4 | 14 | 2 | 14 | 3000 | 4.2 | 35 | 40 |
| 5 | Muar | 1580 | 5.3 | 100 | 2.1 | 30 | 4000 | 5.8 | 55 | 45 |
| 6 | Endau | 1682 | 5 | 200 | 2.5 | 50 | 6800 | 6.5 | 72 | 45 |
| 7* | Maputo | 4550 | 2.3 | 16 | 2.3 | 16 | 5000 | 3.9 | 100 | 70 |
| 8* | Thames | 67000 | 21 | 21 | 21 | 21 | 0 | 9.7 | 50 | 51 |
| 9 | Corantijn | 26670 | 19 | 60 | 8 | 60 | 18000 | 6.8 | 400 | 40 |
| 10* | Sinnamary | 1155 | 2.8 | 40 | 1.5 | 21 | 3000 | 3.6 | 95 | 50 |
| 11 | Mae Klong | 1038 | 1.8 | 200 | 1.8 | 300 | 3400 | 5.2 | 150 | 40 |
| 12* | Lalang | 3184 | 90 | 90 | 49 | 49 | 0 | 7.1 | 130 | 70 |
| 13* | Limpopo | 1075 | 50 | 200 | 18 | 200 | 22000 | 7.1 | 90 | 43 |
| 14* | Tha Chin | 1430 | 2.2 | 80 | 2.2 | 80 | 5800 | 5.5 | 45 | 50 |
| 15* | Chao Phraya | 3508 | 100 | 100 | 26 | 300 | 18000 | 8.5 | 200 | 51 |
| 16 | Edisto | 5401 | 2.1 | 16 | 2.1 | 23 | 2000 | 4.0 | 60 | 30 |
| 17* | Elbe | 25472 | 29 | 90 | 19 | 90 | 36000 | 9.4 | 350 | 43 |
| 18* | Shatt al-Arab | 4260 | 22 | 160 | 26 | 230 | 14000 | 8.0 | 250 | 38 |

Note (a): The estuaries with asterisk-marked label used $K_m$ from Cai et al. (2012), while others from Gisen (2015a).

Note (b): Data about Shatt al-Arab Estuary comes from Abdullah et al. (2016).



**Table 2.** Summary of salinity measurement

| Label | Date | $s_1$ [psu] | $E_1$ [m] | $L$ [m] | $T$ [s] | $Q_f$ [m³/s] | $\delta_H$ $(10^{-6})[\text{m}^{-1}]$ | $H_1$ [m] | $\eta/h$ [%] |
|---|---|---|---|---|---|---|---|---|---|
| 1* | 28-02-2013 | 15 | 9189 | 11000 | 44400 | 50 | -6.3 | 2.8 | 25 |
| 2* | 13-03-2013 | 8 | 12651 | 16000 | 44400 | 316 | 3 | 2.5 | 28 |
| 3* | 21-06-2012 | 28 | 14103 | 42000 | 44400 | 42 | 1.7 | 3.5 | 50 |
| 4* | 24-07-2012 | 14 | 12560 | 14000 | 44400 | 42 | -3.7 | 4.0 | 47 |
| 5* | 03-08-2012 | 18 | 10883 | 35000 | 44400 | 35 | -2.68 | 2.0 | 17 |
| 6* | 28-03-2013 | 17 | 10408 | 21000 | 44400 | 54 | -1.3 | 1.9 | 14 |
| 7a | 28-04-1982 | 29 | 13131 | 21000 | 44440 | 25 | 2 | 3.5 | 45 |
| 7b | 15-07-1982 | 32 | 8080 | 30000 | 44440 | 8 | 2 | 2.1 | 27 |
| 7c* | 19-04-1984 | 22 | 13131 | 20000 | 44440 | 120 | 2 | 3.3 | 43 |
| 7d | 17-05-1984 | 24 | 13131 | 20000 | 44440 | 50 | 2 | 3.4 | 44 |
| 7e | 29-05-1984 | 26 | 12626 | 23000 | 44440 | 40 | 2 | 3.0 | 39 |
| 8* | 07-04-1949 | 31 | 14000 | 83000 | 44400 | 40 | 1.1 | 5.3 | 27 |
| 9a* | 09-12-1978 | 14 | 11638 | 58000 | 44440 | 120 | -1.7 | 1.8 | 13 |
| 9b | 14-12-1978 | 12 | 12608 | 63000 | 44440 | 130 | -1.7 | 2.2 | 16 |
| 9c | 20-12-1978 | 10 | 12608 | 58000 | 44440 | 220 | -1.7 | 1.6 | 11 |
| 10a | 12-11-1993 | 9 | 8472 | 7600 | 44440 | 168 | -5 | 2.6 | 36 |
| 10b | 27-04-1994 | 7 | 10836 | 7800 | 44440 | 148 | -5 | 2.9 | 40 |
| 10c* | 03-11-1994 | 12 | 9851 | 9600 | 44440 | 112 | -5 | 2.9 | 40 |
| 11a* | 08-03-1977 | 24 | 9858 | 23000 | 44400 | 60 | -4.2 | 1.5 | 14 |
| 11b | 09-04-1977 | 25 | 7886 | 28000 | 44400 | 12 | -4.2 | 2.1 | 20 |
| 12* | 20-10-1989 | 14 | 29000 | 18000 | 86400 | 120 | -0.54 | 2.6 | 18 |
| 13a | 31-12-1982 | 24 | 7267 | 67000 | 44440 | 2 | 1.7 | 1.1 | 8 |
| 13b | 14-07-1994 | 12 | 7267 | 47000 | 44440 | 5 | 1.7 | 1.0 | 7 |
| 13c* | 24-07-1994 | 15 | 8305 | 58000 | 44440 | 5 | 1.7 | 0.93 | 7 |
| 13d | 10-08-1994 | 17 | 8305 | 62000 | 44440 | 3 | 1.7 | 1.0 | 7 |
| 14a* | 27-02-1986 | 21 | 18807 | 37000 | 44400 | 40 | -10.6 | 2.4 | 22 |
| 14b | 01-03-1986 | 25 | 13560 | 42000 | 86400 | 40 | -5.5 | 1.8 | 17 |
| 14c | 13-08-1987 | 17 | 11284 | 34000 | 44400 | 39 | -10.6 | 1.9 | 17 |
| 15a* | 05-06-1962 | 11 | 23068 | 43000 | 86400 | 63 | -2.2 | 2.1 | 12 |



| Label | Date | $s_1$ [psu] | $E_1$ [m] | $L$ [m] | $T$ [s] | $Q_f$ [m³/s] | $\delta_H$ $(10^{-6})$[m⁻¹] | $H_1$ [m] | $\eta/h$ [%] |
|---|---|---|---|---|---|---|---|---|---|
| 15b | 16-01-1987 | 1 | 13456 | 22000 | 86400 | 180 | -2.2 | 2.4 | 14 |
| 15c | 23-02-1983 | 8.5 | 18262 | 38000 | 86400 | 100 | -2.2 | 1.5 | 9 |
| 15d | 29-01-1983 | 12 | 24991 | 44000 | 86400 | 90 | -2.2 | 1.5 | 9 |
| 16a⋆ | 12-07-2010 | 50 | 12773 | 35000 | 44400 | 15 | -8.8 | 2.3 | 28 |
| 16b | 13-07-2010 | 48 | 12773 | 38000 | 44400 | 14 | -8.8 | 2.3 | 28 |
| 16c | 14-07-2010 | 48 | 12282 | 37000 | 44400 | 25 | -8.8 | 2.3 | 28 |
| 16d | 15-07-2010 | 50 | 12282 | 35000 | 44400 | 25 | -8.8 | 2.3 | 28 |
| 17a⋆ | 21-09-2004 | 10 | 21493 | 68000 | 44440 | 200 | 2 | 2.2 | 11 |
| 17b | 21-09-2004 | 10.5 | 19344 | 69000 | 44440 | 200 | 2 | 3.2 | 17 |
| 18a | 26-03-2014 | 11 | 9324 | 40000 | 44000 | 114 | -5 | 1.6 | 10 |
| 18b | 16-05-2014 | 15 | 9324 | 48000 | 44000 | 96 | -5 | 2.3 | 15 |
| 18c | 24-09-2014 | 27 | 14452 | 65000 | 44000 | 58 | -5 | 2.2 | 14 |
| 18d⋆ | 05-01-2015 | 15 | 9324 | 42000 | 44000 | 63 | -5 | 2.4 | 15 |

Note: The data chosen from each estuary with star-marked label is used for empirical calibration.





**Table 3.** Dispersion parameters using $C_2 = 10$

| Label | $K$ [-] | $D_1$ [m$^2$/s] | $\alpha$ [m$^{-1}$] | $\beta$ [-] | $N_R$ [-] | $K_{\mathrm{calculated}}$ [-] |
|---|---|---|---|---|---|---|
| 1 | 0.78 | 370 | 7.4 | 9.4 | 0.55 | 0.51 |
| 2 | 0.54 | 255 | 0.81 | 3.3 | 0.041 | 0.51 |
| 3 | 0.49 | 234 | 5.6 | 0.49 | 0.022 | 0.52 |
| 4 | 0.51 | 314 | 7.5 | 0.94 | 0.084 | 0.51 |
| 5 | 0.45 | 326 | 9.3 | 3.1 | 0.12 | 0.52 |
| 6 | 0.65 | 282 | 5.2 | 15 | 0.23 | 0.53 |
| 7a | 0.70 | 80 | 3.2 | 0.77 | 0.019 | 0.57 |
| 7b | 0.69 | 47 | 5.9 | 0.41 | 0.028 | 0.62 |
| 7c | 0.57 | 281 | 2.3 | 0.86 | 0.068 | 0.52 |
| 7d | 0.65 | 135 | 2.7 | 0.85 | 0.031 | 0.54 |
| 7e | 0.63 | 133 | 3.3 | 0.67 | 0.030 | 0.55 |
| 8 | 0.55 | 239 | 6.0 | 0.030 | 0.0044 | 0.65 |
| 9a | 0.61 | 178 | 1.5 | 0.92 | 0.018 | 0.55 |
| 9b | 0.55 | 206 | 1.6 | 0.78 | 0.014 | 0.53 |
| 9c | 0.51 | 292 | 1.3 | 0.86 | 0.019 | 0.52 |
| 10a | 0.52 | 368 | 2.2 | 8.2 | 0.53 | 0.51 |
| 10b | 0.52 | 335 | 2.3 | 8.0 | 0.17 | 0.51 |
| 10c | 0.54 | 359 | 3.2 | 5.8 | 0.30 | 0.51 |
| 11a | 0.52 | 484 | 8.1 | 12 | 0.51 | 0.51 |
| 11b | 0.58 | 177 | 15 | 7.6 | 0.21 | 0.54 |
| 12 | 0.74 | 456 | 3.8 | 5.5 | 0.077 | 0.52 |
| 13a | 0.72 | 46 | 23 | 5.8 | 0.056 | 0.62 |
| 13b | 0.67 | 63 | 13 | 10 | 0.070 | 0.58 |
| 13c | 0.61 | 86 | 17 | 6.6 | 0.059 | 0.57 |
| 13d | 0.64 | 62 | 21 | 5.8 | 0.040 | 0.59 |
| 14a | 0.45 | 536 | 13 | 1.9 | 0.033 | 0.51 |
| 14b | 0.45 | 592 | 15 | 1.7 | 0.77 | 0.52 |
| 14c | 0.45 | 376 | 9.6 | 2.6 | 0.12 | 0.51 |
| 15a | 0.65 | 336 | 5.3 | 3.5 | 0.068 | 0.53 |





| Label | $K$ | $D_1$ | $\alpha$ | $\beta$ | $N_R$ | $K_{\text{calculated}}$ |
| | [-] | [m$^2$/s] | [m$^{-1}$] | [-] | [-] | [-] |
|---|---|---|---|---|---|---|
| 15b | 0.58 | 163 | 0.90 | 18 | 0.089 | 0.51 |
| 15c | 0.62 | 402 | 4.0 | 4.4 | 0.17 | 0.53 |
| 15d | 0.62 | 485 | 5.4 | 3.3 | 0.083 | 0.52 |
| 16a | 0.58 | 122 | 8.1 | 0.21 | 0.018 | 0.56 |
| 16b | 0.55 | 130 | 9.3 | 0.18 | 0.016 | 0.56 |
| 16c | 0.49 | 219 | 8.8 | 0.16 | 0.033 | 0.54 |
| 16d | 0.53 | 195 | 7.8 | 0.20 | 0.034 | 0.54 |
| 17a | 0.62 | 142 | 0.71 | 3.1 | 0.0050 | 0.53 |
| 17b | 0.62 | 149 | 0.74 | 2.9 | 0.0073 | 0.53 |
| 18a | 0.48 | 290 | 2.5 | 7.1 | 0.19 | 0.52 |
| 18b | 0.45 | 324 | 3.4 | 5.0 | 0.22 | 0.52 |
| 18c | 0.54 | 403 | 6.9 | 2.9 | 0.064 | 0.53 |
| 18d | 0.52 | 234 | 3.7 | 5.2 | 0.14 | 0.54 |