# Peer review of "The physics behind Van der Burgh's empirical equation, providing a new predictive equation for salinity intrusion in estuaries"

_Hydrology and Earth System Sciences, 2016_

## Referee Comment (RC1) · Anonymous Referee #1 · 4 Feb 2017

The author gives an emphasis on the physical foundation of Van der Burgh's equation, K, that makes a connection between the empirical equation and theoretical literature, which limits K value between 0.5 and 0.66 keeping within the feasible range ($0 < K < 1$). Additionally, a one-dimensional predictive equation has been developed for salinity dispersion along the estuary which subsequently provide a new predictive equation for the longitudinal salinity distribution. Finally this equation has applied to 18 estuaries among which the three estuaries have the constant depth. This paper can attract a great deal of attention. However the manuscript has critical issues in applying of the model to the real estuary.

[Figure]

Major comments:

1. This new equation limits K value between 0.5 and 0.66. If K is 0.5, then density-driven dispersion occurs and dispersion is proportional to salinity gradient (page 10, line 24-25). If K is 0.66, dispersion is dominated by lateral exchange in wider part of estuaries and proportion to the square of salinity gradient (Page 10, line25-26). In addition, the calibrated K values of 18 estuaries ranged from 0.45 to 0.78. Does it mean only density-driven or lateral exchange in the 18 estuaries? New K suggests that dispersion is dominated by density-driven in most estuaries where were classified as tide-driven dispersion (see example in Table 1). What happen in real estuaries? This should be addressed.

2. What sort of mechanisms are responsible for dispersion downstream of Maputo (10 km from the mouth, Fig. 4), Kurau (5 km from the mouth, Fig. 4) and Endau (10 km from the mouth, Appendix C) where there is no horizontal salinity gradient? In addition, the larger the K value, the stronger the lateral exchange (Page 10, line 28-29). Previous literature (Savenije, 2005, 2012) implies that if K is larger, the stronger gravitational circulation exist near the toe of salt intrusion. Is it contradict with the previous literature? Moreover, the new equation narrowing the range of K, and makes contradiction with the K value of some estuaries (Table 5.4, 5.6 Savenije, 2012).

3. The author suggested the constant K value spatially if the depth is constant (Line 22, page 10: "yet, from Eq. (10), the K value should be constant if the depth is constant"). Among 18 estuaries where the newly developed Van der Burgh's equation has been applied to, only three estuaries (7, 8, and 14, Table 1) have the constant depth as the values of a and b are the same (line 3, page 9, Table 1). Moreover, K is affected by tide, geometry (Gisen, 2015) and freshwater discharge (Shaha and Cho, 2011; this study also) (line 19 page 10). What is the justification to suggest the constant K for an estuary? This would be better to explain for understanding clearly by the reader.

Minor comments:

1. The tidal excursion length E appears first in Eq. (10). This would better to define here rather than after Eq. (12). 2. In line 5, page 6: The lateral exchange is proportional to longitudinal (Fischer, 1972). This should be longitudinal salinity gradient. 3. In line 3, page 10: "A" smaller value. "A" should be small letter. 4. Figure 2: S1, S2, S3, SL and SR are not clear. 5. Table 3: Kcalibrated would be better instead of K.

[Figure]

Table 1. Comparison of calibrated K values of previous (Savenije, 2012) and new studies that represents the different dispersion

| Estuaries | K values (**Previous**) | Type of dispersion (**Previous**) | K values (**this study**) | Type of dispersion (**new calculation\*\***) |
|---|---|---|---|---|
| Maputo | 0.38 | Tide-driven | 0.57-0.70 | density-driven |
| Thames | 0.2 | Tide-driven | 0.55 | density-driven |
| Corantijn | 0.21 | Tide-driven | 0.51-0.61 | density-driven |
| Mae Klong | 0.3 | Tide-driven | 0.52-0.58 | density-driven |

**Fig. 1.**

---

## Referee Comment (RC2) · Anonymous Referee #2 · 14 Mar 2017

Van der Burgh's equation is a widely used formulation for the saline water intrusion in alluvial estuaries. While the simple one-dimensional equation has be shown to be sufficiently accurate to predict the salinity distribution, the physical explanation suffers. The authors therefor physically explain the equation for an parameter range $1/2 < K < 2/3$. The theoretical derivation ends with an analytical expression for the Van der Burgh's coefficient K. The authors conclude that their comparison with a theoretical study gives a solid foundation for the use of K in one-dimensional models.

In the second part of the study the authors provide an analytical solution for the saline

water intrusion additional including the residual circulation, if the estuary is wide. In the last section the authors validate their formulations by testing them with 18 different estuaries.

Major comments

1) The authors give a solid explanation for a physical range of the K value ($1/2 < K < 2/3$). Therefore I would expect a substantially critical handling with the values achieved by the calibration ($0.45 < K < 0.78$). For predictive use in real applications uncertainties of about 16 % could not be seen as "quite similar". The reader of the study will not be sufficiently aware of possible uncertainties in the application of the formulations. What could be reasons for the uncertainties? What are the uncertainties of calibration process itself? Which physical expressions are still missing in the estimations? 2) In figure 5 a 1:1 plot is presented. The plot shows not only a dot of white noise, but something like a linear correlation between predicted and calibrated K. This correlation allows the assumption that a systematic underestimation occurs in the prediction of K. What could be reason for that bias? Which terms should be considered in future work? In the same figure 5 three estuaries (1, 8, 12) can be identified as outliers in the relation between predicted and calibrated K. What is different in these estuaries that the relation between predicted and calibrated K differs to the other 15 estuaries. 3) In the second part of the study the authors include the effect of the residual circulation; the strength of the influence is controlled by the coefficient C2. The results shown in appendix C do not convince me. I cannot find 14 estuaries which "perform perfectly". Here a more critical view on the complex three-dimensional and also non-linear terms in equations explaining the mixing and straining processes in the mouth regions of estuaries should be given.

---

## Author Comment (AC1) · 12 Apr 2017

Zhilin Zhang and Hubert H. G. Savenije

z.zhang-5@tudelft.nl

We would like to thank referee #1 for his/her insightful comments and the positive feedback, which we have received. It has given us quite a lot to reflect on, which is why our reply is rather late. Below we reply to the major comments in detail. The minor comments will all be addressed in the final version. We thank you for the detailed reading. Major comments 1. Regarding your first observation on the discrepancy between the theoretical and the empirical values of the Van der Burgh coefficient. The Taylor series to approximate equation (8) is not bound by K=[1/2, 2/3]. If we consider (-D ds/dx) as

the exchange flow, with K=0, 1/2, 2/3, 3/4, ... then this exchange flow is proportional to (ds/dx), (ds/dx)2, (ds/dx)3, (ds/dx)4, ... Therefore, if K=0, the dispersion coefficient is not dependent on the salinity gradient (completely tide-driven) and the larger the K value (K is bound between [0,1] (Savenije, 2005)), the more sensitive the dispersion is to the salinity gradient. In order to link Van der Burgh's empirical coefficient to physical parameters, we compared the Taylor series equation (9) to equation (7), but only for the density-driven part. In the theoretical derivation, the K then has two extreme values: 1/2 and 2/3. According to equation (10), K is determined by the fresh water discharge and tidal excursion (these two parameters also affect the stratification number). In a case of a relatively constant discharge, a larger tidal excursion implies less stratification (well-mixed situation) and K approaching the lower limit. On the other hand, a smaller tidal excursion implies more stratification and K approaching the higher limit, which corresponds to the situation where the dispersion is more sensitive to the salinity gradient. Although MacCready's density-driven dispersion limited the range of K to [1/2, 2/3], in reality this range can be larger. Fully tide-driven dispersion would correspond with K=0, but stronger 3-D density-driven mixing (including lateral density-driven exchange flows) could correspond with K-values larger than 2/3. Depending on the importance of tide-driven mixing and these 3-D effects, the full range of [0,1] is feasible. The comparison made in the paper, however, only applies to vertical 2-D density-driven dispersion.

During the empirical calibration using the new predictive equation, in principle, all mechanisms are included, leading to K values falling within the entire feasible range of 0<K<1. Moreover, according to equation (4), the K value affects the salinity most in the upstream reach, where D/D1 is small. When the salinity gradient is relatively large, the calibrated K value is large suggesting predominantly density-driven dispersion (e.g., Kurau estuary), and when the gradient is small, the K is small suggesting predominantly tide-driven dispersion (e.g., Muar estuary).

In this paper, the authors tried to link these two theoretical and empirical approaches to explore the physics behind Van der Burgh's empirical equation. Even though the

theoretical derivation was limited to the range of (1/2, 2/3) and the Taylor series has a larger range, the authors expected a correspondence between the two methods.

The calibrated K for real estuaries ranged from 0.45 to 0.78, while 34 out of 42 values were in between 1/2 and 2/3. This difference can be explained. First of all, there is quite some uncertainty in calibrating a partly empirical analytical model to data in real estuaries, as a result of a whole range of uncertain factors related with observational errors, data problems, the assumption of steady state, and many other factors. This alone can lead to a substantially larger spread in the values obtained than would be expected from a purely theoretical derivation (which also has its limiting assumptions). But if for the sake of the argument it is assumed that the calibrated values that fall outside the range of (1/2, 2/3) have a physical explanation, then a value smaller than 1/2 would imply that tide-driven mixing plays a prominent role in the estuary. On the other side of the spectrum, with a larger dS/dx, the estuary would be more stratified. Moreover, the theoretical approach follows width-averaged dynamics whereas the empirical approach relies on natural convergent estuaries. A K value larger than 2/3 could result from a strong lateral salinity gradient due to shearing in a complex geometry, which strengthens the sensitivity to the salinity gradient (as observed by Fischer, 1972). But looking at the calibrated range, most estuaries have a K value in the range, say (0.5, 0.6), suggesting that most of the estuaries are more well-mixed in the upstream part.

Regarding the observation that we obtained different K-values compared to previous studies, we note that in previous studies the K values were calibrated using a different analytical equation. The traditional equation used a boundary condition at the mouth (D0) and excluded the residual circulation term. The equations were then integrated considering constant depth and no damping. In the present research, dispersion is calculated locally and K values can vary. This is the reason why earlier studies arrived at different values compared to the ones obtained in this paper. In view of this, we intend to change the description in Table 1. The estuaries that have low K values (relatively close to 0.5) are in fact predominantly tide-driven and well-mixed.

2. Regarding the question of what is happening in the most downstream parts of the Kurau, Endau and Maputo. In the wider parts of these estuaries (the mouth of the trumpet), we generally see clear ebb and flood shears. Whether these ebb and flood channels are the result of the widening or the cause of the widening, is something to be investigated, but a fact is that they are there. We clearly see pronounced ebb and flood channels where relatively saline water enters the flood channel and relatively fresh water leaves the ebb channel. These channels can have lengths in the order of the tidal excursion and can transport saline water deep into the estuary. At the cross-over points fresh water carried from upstream through an ebb channel meets saline water carried from downstream through a flood channel, which causes substantial tide-driven mixing, even though longitudinal salinity gradients are small near the mouth. This mechanism of ebb/flood channel tide-driven residual circulation was described by Nguyen et al. (2008). By adding the additional term in the equation we cater for this important mechanism in wide estuary mouths.

The findings in this paper (the larger the K values, the stronger the stratification) are in line with earlier work, in that the larger the K value, the stronger the gravitational circulation. Regarding the narrowing of the range, we think that the 2-D vertical theoretical approach is too limited and that K can be larger than 2/3; real estuaries have a larger range, as shown by the calibration.

3. The authors have indeed abandoned the idea that K is constant in space (and time). However, for calibration we have to use a spatially constant value. Fortunately, according to the equations (10) and (11), the K value does not vary that much in space since the freshwater discharge is unchanged spatially and the changes of estuarine depth as well as tidal damping/amplification are gradual and relatively small. Over time, however, K can vary substantially as a result of different discharges and tidal amplitude.

---

## Author Comment (AC2) · 12 Apr 2017

We would like to thank referee #2 for his/her detailed reading and valuable comments. It triggered us to further consider the results obtained and it helped us to formulate better what precisely the implication is of a larger range of Van der Burgh's K. As a result of your comments we shall modify the paper, particularly the discussion of the results obtained.

1. The first comment of Referee #2 is very similar to the one made by Referee #1. We replied to it there in much detail. In the revised paper, we shall try to estimate what

the uncertainties are that may have affected the wider range of K values (observational errors, schematization errors, underlying assumptions, and the fact that the model is for steady state, whereas an estuary is never in steady state). Also we should realize that the mixing length used in our empirical model is the tidal excursion (appropriate for well-mixed estuaries) while for more stratified estuaries, the mixing length is the depth. The authors used the tidal excursion because it appears that tide-driven mixing is dominant in well-mixed estuaries (e.g., Wei et al., 2016) and we assumed that most estuaries achieve a more well-mixed condition, especially during spring tide and low flow when the salinity intrusion becomes an issue. Hence, the correct mixing length to use (E or h) may depend on the stratification, which the authors will study further.

2. We have redrawn Figure 5 taking into account 25 % sensitivity of fresh water discharge (x-err) and calibration errors (y-err).

After inclusion of these uncertainty bounds, we can still see that the predictive method overestimates the low calibrated K values, and underestimates the high values. Or, that the range of (1/2, 2/3) appears too narrow. The reasons are that the predictive method does not consider the tide-driven mixing (which draws the predicted K-value further down) and that for high K-values the 2-D vertical theoretical approach underestimates lateral density driven mixing (which would increase the high values), as discussed in the reply to reviewer #1. So in estuaries with a low predicted K-value (which do not have much stratification) the K value should be even smaller as a result of tidal mixing; in estuaries with a high predicted K-value (which do have substantial stratification) the K value should be even larger due to complex 3-D density-driven circulation. In the central reach, the prediction by MacCready's equation is not bad.

In addition, there are some outliers. The outliers in estuaries #1 and #12 (Kurau and Lalang), which have very high calibrated K values, density effects may be so strong that they cause three-dimensional gravitational circulation, or the fresh water discharge may have been substantially underestimated. On the other hand in the Thames estuary (#8) tide-driven dispersion, such as tidal trapping, may have considerable influence.

3. In wide estuaries, strong residual circulation develops as a result of separate flows through the preferential ebb and flood channels that meet at cross-over points where fresh water from upstream mixes with saline water from downstream. The length of such channels is in the order of the tidal excursion. We introduced this additional tide-driven mechanism in the wider part of the estuaries by an introduction of an additional lateral dispersion factor of (1+C2(B/E)2). This factor adds lateral exchange flow and enlarges the value of the dispersion, leading to a bending down of the simulated salinity curves near the (wide) mouth. When C2=10, the salinity curves (14 out of 18 estuaries) bend down perfectly fitting the observations. Possible reasons for these outliers as well as the poor fit in the downstream parts of the Lalang and Chao Phraya have been discussed in Section 4.2 of the paper. So instead of using a different C2 factor for each estuary, a constant value has been used so as to make the calibration simpler and more consistent.

Reference:

Wei, X., Schramkowski, G.P., Schuttelaars, H.M., 2016. Salt dynamics in well-mixed estuaries: importance of advection by tides. Journal of Physical Oceanography, 46(5), 1457-1475.
* * *
[Figure]

**Fig. 1.** Figure 5